# Using Optical Tracking System Data to Measure Team Synergic Behavior: Synchronization of Player-Ball-Goal Angles in a Football Match

**DOI:** 10.3390/s20174990

**Published:** 2020-09-03

**Authors:** Daniel Carrilho, Micael Santos Couceiro, João Brito, Pedro Figueiredo, Rui J. Lopes, Duarte Araújo

**Affiliations:** 1CIPER, Faculty of Human Kinetics, University of Lisbon, 1495-751 Cruz Quebrada, Portugal; micael@ingeniarius.pt (M.S.C.); daraujo@fmh.ulisboa.pt (D.A.); 2Ingeniarius, Lda., 3025-307 Coimbra, Portugal; 3Portugal Football School, Portuguese Football Federation (FPF), 1495-433 Cruz Quebrada, Portugal; joao.brito@fpf.pt (J.B.); pedro.figueiredo@fpf.pt (P.F.); 4Research Centre in Sports Sciences, Health Sciences and Human Development, CIDESD, University Institute of Maia, ISMAI, 4475-690 Maia, Portugal; 5Department of Science and Information Technology, ISCTE, University Institute of Lisbon, 1649-026 Lisbon, Portugal; rui.lopes@iscte-iul.pt; 6Telecommunications Institute, ISCTE, University Institute of Lisbon, 1049-001 Lisbon, Portugal

**Keywords:** spatiotemporal data, synergies, cluster phase analysis, order parameter, team behavior, group kinematics, football analytics, ecological dynamics, soccer

## Abstract

The ecological dynamics approach to interpersonal relationships provides theoretical support to the use of kinematic data, obtained with sensor-based systems, in which players of a team are linked mainly by information from the performance environment. Our goal was to capture the properties of synergic behavior in football, using spatiotemporal data from one match of the 2018 FIFA WORLD CUP RUSSIA, to explore the application of player-ball-goal angles in cluster phase analysis. Linear mixed effects models were used to test the statistical significance of different effects, such as: team, half(-time), role and pitch zones. Results showed that the cluster phase values (synchronization) for the home team, had a $3.812\times 10^{-2}\pm 0.536\times 10^{-2}$ increase with respect to the away team ($X^{2}\left(41\right)=259.8,\;p<0.001$) and that changing the role from with ball to without ball increased synchronization by $16.715\times 10^{-2}\pm 0.283\times 10^{-2}$ ($X^{2}\left(41\right)=12227.0,\;p<0.001$). The interaction between effects was also significant. The player-team relative phase, the player-ball-goal angles relative frequency and the team configurations, showed that variations of synchronization might indicate critical performance changes (ball possession changes, goals scored, etc.). This study captured the ongoing player-environment link and the properties of team synergic behavior, supporting the use of sensor-based data computations in the development of relevant indicators for tactical analysis in sports.

## 1. Introduction

### 1.1. Tracking Data in Football

The use of electronic performance-tracking systems to obtain spatiotemporal data in association football (football, for simplicity) is becoming a generalized practice for performance analysis. These systems include global positioning systems, radio-based local positioning systems and video camera-based systems, which provide big data about the individual performances of players and tactical behaviors of teams [1,2,3].

Optical-based camera systems have the advantage of being non-invasive to players, using multi-camera units and monocular systems around the pitch. These systems are able to provide high rate and high spatial resolution data from the position of the players and the ball, opening a door for the increasing development of football analytics [1,4]. Football clubs and national teams are increasingly investing in sports analytics departments, specialized in processing the data, computing compound variables and performance indicators, and visually presenting them to coaches, players and others.

Although these new opportunities are very important for the development of players and teams in an increasingly competitive context, they are inherently bottom-up approaches, emerging from the data. Thus, they can be complemented by a theoretically informed approach, such as the ecological dynamics, that can contextualize and explain the phenomenon from where data were obtained.

### 1.2. Synergies in Sports

The study of sports teams, from an ecological dynamics theoretical framework, is based on the understanding of how players collectively and in coordination behave in a football match, detecting patterns of information (e.g., movements from the opponent team) in the specific performance environment [5]. There is evidence that the main intra-team communication channel is the perception of shared affordances, understood as opportunities for action offered by the performance environment and perceived by trained groups of players, that evolve over time [6,7].

From an ecological dynamics perspective, compound variables called order parameters, capture collective behavior, in this case team behavior [8,9]. By collectively capturing the team behavior dynamics, the influences of local changes (e.g., inter-team distance) can be expressed in the values of order parameters (e.g., changing from collective attack to defense), indicating the emergence of functional relations between the elements within the team and the specific environment [10,11].

The inseparable link between the players and the performance environment is a fundamental and contemporary topic when studying kinematics in football (i.e., the movement of players and teams), which is being increasingly related to the concept of interpersonal synergies [5,12]. The concept of synergies was coined by Bernstein, to functionally and structurally describe coordinative movement [13]. However, there is still some ambiguity in its definition, which is a consequence of defining synergies as hard-wired or soft-assembled [14]. The ecological dynamics approach focuses on the functional characteristic of synergies, which emerge and dissipate according to task constraints. While other approaches that also embrace variability, such as the uncontrolled manifold approach, aim to quantify the variance of elemental variables that allow for desired levels of performance to be maintained, the goal of the ecological dynamics approach is to analyze the dynamics of self-organization of order parameters and other related properties of synergies [5,15,16]. Specifically, in team sports, team synergies imply that players are perceptually attuned to shared affordances, provided by events in the match, to which players respond collectively as a coherent unit, a synergy [5].

The ecological dynamics approach provides the theoretical guidance for an otherwise methodological practice of data analysis in the sport. This is particularly relevant in a time when sensor-based systems are providing data that have descriptive and predictive value, and can also have explanatory value.

### 1.3. The Properties of Team Synergies in Sports

Team synergic behavior has properties that can be identified across different tasks, such as: Dimensional Compression, Reciprocal Compensation, Interpersonal Linkages and Degeneracy [5]. In this section, each property is briefly described and examples are provided about how data from sensor-based systems were used to compute each one of them.

Dimensional compression is defined as the representation of collective behavior by a low-dimensional variable, an order parameter. This property captures, in one variable, the behavior of a team computed from positional kinematic data, in quantities such as synchronization among players (cluster phase or relative phase analysis), centroid of a team, stretch index or team space occupation, captured by Voronoi cells [17,18,19,20,21,22,23].

Reciprocal compensation is captured by the adjustments of the players of a team, indicating how teammates compensate each other’s positioning, maintaining the stability of the order parameter [5]. Therefore, understanding these variations enables the analysis of the environmental conditions in which the order parameter is stabilized and at what conditions it loses its stability. For example, readjustment delay was used to measure this property in football, as the authors aimed to measure the positional adaptation of footballers in relation to their teammates [24].

Interpersonal linkages can be understood as the contribution of each individual to the team synergy. This property enhances the importance of taking into account the interpersonal variability of the elements within a team. In comparison to the study of collective behavior in other biological systems, where agents are somehow considered as equivalents [25,26,27], this property “highlights the need to consider that each element is unique, and this implies an understanding about team behavior that is different from considering a team as a superorganism” [5] (p. 8). To capture this property, tools such as positional heatmaps or Voronoi tessellations have been applied [20,21]. More recently, dynamic variables have been included to calculate space control, such as the speed and direction of movement and the value of on-ball and off-ball actions [28,29,30,31,32].

Degeneracy describes the adaptable and flexible behavior of the team, expressing how players in coordination adjust to the momentary (e.g., opposition behavior) or more persistent (e.g., substitution of a player) changes in the performance environment. The players within the team are adaptable, even though they have specific characteristics or roles, which guarantees that the team as a whole can adapt to local changes. “Understanding synergies is far more than identifying and understanding the functional structure of each individual synergy” [5] (p. 9). Synergies form, transform and dissipate according to the momentary characteristics of the context and different synergies can be formed for the same task, enhancing the need to understand synergies at different levels of organization. Social network analysis can capture this degenerate behavior in sports. Studies have been using these metrics to demonstrate local structures of organization and identify specific characteristics of each network, such as centrality, density or preferable circulation pathways [33]. More recent approaches have extended this idea and used hypernetworks to identify coordination and organization dynamics within the team, at several levels of organization [34,35,36].

### 1.4. Cluster Phase Analysis

Relative phase analysis (RPA) has been used in team sports to measure synchronization [37]. RPA is based on the difference of the oscillation between two phases, expressed as a measure of their relative phase angle, defined by their angular frequency (ω) and initial phase (ϕ0) [22]. Two oscillators in a time series can be locked in an in-phase mode of synchronization, if their angle difference holds near 0° or in an anti-phase, if the angle is near 180° [22]. Based on this idea, cluster phase analysis (CPA) also measures synchronization, but it is based on the relationship among the oscillatory movement of a group of elements, using the Kuramoto order parameter [22,38,39]. Therefore, contrary to RPA, CPA measures synchrony among the oscillating phases of the elements within a group. To do that, CPA presupposes that the elements of the system, (i.e., the players in a football team) display an oscillatory behavior. “The phase of an oscillator lies within the interval 0 and 2π in the unit circle and this (the phase) is viewed as being a periodic variable that can be represented in the complex plane” [22] (p. 101)”.

Previous applications of RPA and CPA in team sports made use of the Hilbert transform (HT) to obtain the oscillation angles [18,24]. However, in a recent application of CPA to football, López-Felip and colleagues raised important issues associated with applying the HT to these types of data and highlighted the need to measure synchronization taking into account specific variables of the performance environment [22]. Specifically, the main issue identified was that, when applied to football analysis, a pre-requisite of the HT is to separate longitudinal from lateral displacement data, to obtain the continuous phase angles. However, the authors pertinently observed that “…visually guided behavior in soccer does not consist in coupling continuous phase angles, but in informational couplings between teammates and opponents, as well as relevant properties from the joint-action task space (i.e., limits of the field, goals, etc.)” [22] (p. 100).

In the present study, we aimed to explore the application of player-ball-goal angles (PBGA) in CPA to directly measure player–environment relationships and capture the properties of team synergic behavior, using kinematic, sensor-based data. The ball, the goals and the players were identified as key ecological variables of the football match, as they inform and constrain the behavior of players. Therefore, by using the PBGA we aimed to capture the link between players and performance environment, providing ecological explanatory value to CPA. The synchronization measures obtained from CPA, were a direct result of changes in the PBGA, which continually captured the displacement of the players on the pitch, in relation to the goal and the ball.

## 2. Methodology

### 2.1. Kinematic Spatiotemporal Data

Spatiotemporal positional data were used from one match of the 2018 FIFA WORLD CUP RUSSIA. Data were provided by the Portuguese Football Federation (FPF) and obtained by the TRACAB Optical Tracking System (ChyronHego), which was recently validated and is acknowledged as one of the market leaders in the field [1]. The dataset consisted of positional data of every player (n = 28) and the ball (n = 1), captured at 25 Hz with spatial resolution of 0.01 m. For CPA measures, each cluster (team) was constituted by the players on the pitch (n = 11). There were no players dismissed from the match so the number of players, for both teams, remained the same during the entire time. The dataset provided by this optical tracking system, apart from the planar coordinates, included also the information of which team was in possession of the ball and if the ball was “dead” or “alive” (meaning that the match was running or stopped). This information allowed us to filter the data for the results. Raw data were pre-processed using routines developed in MATLAB R2018b (MathWorks).

### 2.2. Cluster Phase Analysis and Player-Ball-Goal Angles

The PBGA, $\theta_{k}$, for each player, *k*, ranged from 0 to π and was calculated at every time frame, $t_{i}$, with the vertex on the ball (see Figure 1), using the planar coordinates of each player, ball and goal as represented by Equation (1). The PBGA were calculated using the goal being attacked, as we decided that it represented a stronger visual attractor to the players of both teams. In a normal match situation, the team with the ball has the primary objective of attacking the opposition’s goal and the team without the ball has the primary objective to defend its own goal, while trying to recover the ball.
(1)$$\theta_{k}\left(t_{i}\right)=\;atan2(∥P_{k}\left(t_{i}\right)-B\left(t_{i}\right))⊗(G\left(t_{i}\right)-B\left(t_{i}\right)∥\;,\left(P_{k}\left(t_{i}\right)-B\left(t_{i}\right)\right)\cdot \left(G\left(t_{i}\right)-B\left(t_{i}\right))\right)$$

In Equation (1), the PBGA, $\theta_{k}$, was calculated using the planar coordinates for each player, $P_{k}$, ball, B, and goal, G, at every time frame, $t_{i}$, expressed as vectors, namely:$$P_{k}\left(t_{i}\right)=x_{player_{k}}\left(t_{i}\right)\cdot e_{long}+y_{player_{k}}\left(t_{i}\right)\cdot e_{lat}\;B\left(t_{i}\right)=x_{ball}\left(t_{i}\right)\cdot e_{long}+y_{ball}\left(t_{i}\right)\cdot e_{lat}\;G\left(t_{i}\right)=x_{goal}\left(t_{i}\right)\cdot e_{long}+y_{goal}\left(t_{i}\right)\cdot e_{lat}$$

The vectors were calculated using the longitudinal, $e_{long}$**,** and lateral, $e_{lat}$**,** planar coordinates of each player, $x_{player_{k}}$, $y_{player_{k}}$, ball, $x_{ball}$, $y_{ball},$ and goal, $x_{goal}$, $y_{goal}$, at every time frame, $t_{i}$. The PBGA were then submitted to Kuramoto’s order parameter in Equation (2).
(2)$$\overset{´}{r}\left(t_{i}\right)e^{i\psi \left(t_{i}\right)}=\frac{1}{n}\;\overset{n}{\underset{k=1}{\sum }}e^{i\theta_{k}\left(t_{i}\right)}$$

In Equation (2), the coherence of the oscillator population, $\overset{´}{r}$, varying between 0 and 1, was measured using the PBGA, $\theta_{k}$, at every time frame, $t_{i}$ [39]. The average phase is expressed by ψ and n is the total number of elements. In this study, phase coherence or cluster amplitude is referred to as synchronization, considering that values closer to 1 represent a higher level of synchrony between the phase oscillators (players) and closer to 0, a lower level of synchronization.

To capture each one of the properties of team synergic behavior, different elements of CPA were used. For dimensional compression we used the mean and standard deviation (SD) of the synchrony measures and linear mixed effects models to test the statistical significance of interactions between different effects, such as: team, half(-time), role (with ball or without ball) and pitch zones. We divided the pitch in 20 zones as aligned with football terminology when referring to lateral zones (central, central-lateral and lateral) and longitudinal zones (defensive, mid-defensive, mid-offensive and offensive).

For capturing reciprocal compensation, we compared the relative phase of each player to the team with the variations of the synchrony measures of the team. The player-team relative phase, $\Phi_{k}$, was calculated by the difference between each PBGA, $\theta_{k}$, and the group average phase, ψr, at every time frame, $t_{i}$, in Equation (3).
(3)$$\Phi_{k}\left(t_{i}\right)=\;\theta_{k}\left(t_{i}\right)-\;\psi r\left(t_{i}\right)$$

To capture interpersonal linkages, we calculated the relative frequency of the PBGA of every player during the match in the two different roles: with ball and without ball. The relative frequency was calculated, using 18 PGBA bins (±0.174 or 10° in each bin), by adding the number of frames in which each player was in each bin. This allowed us to understand the role of each player within the team, by measuring the relation of each player with the ball and the goal during the match.

Finally, to capture degeneracy we compared different types of team configurations, calculated using three PBGA bins and comparing each team configuration with the group synchrony measures. The amplitude of the PBGA was divided into the following bins, each corresponding to a subgroup: Front Support/Cover (FS/C) = [0, π/3]; Lateral Support/Cover (LS/C) = [π/3, 2π/3]; Back Support/Cover (BS/C) = [2π/3, π]. If the team had possession of the ball, we called the role “support” and if not, we called it “cover”, aligned with football terminology regarding offensive and defensive behaviors.

## 3. Results

We organized the results to show how we were able to capture each synergic property, using the different elements of the CPA equation, based on the PBGA.

### 3.1. Dimensional Compression

Synchronization represented the order parameter that captured the collective behavior of each team in the match. Figure 2 shows the mean and standard deviation (SD) of the synchrony measures, for both teams (away and home), during each half (±45 min) in two ball possession scenarios (roles): with ball and without ball. Next to each measure there is the value of mean synchrony.

Figure 2 shows that the overall synchrony measures for both teams were high (near 1) and that the home team (blue) presented higher mean synchrony values in both halves and roles: first half, with ball (0.79 ± 0.09); first half, without ball (0.84 ± 0.11); second half, with ball (0.82 ± 0.11); second half, without ball (0.83 ± 0.11).

Figure 3 represents the football pitch, divided in 20 zones (5 lateral and 4 longitudinal). The lateral and longitudinal coordinates, that limited each zone, are represented in meters on the *y* and *x* axis, respectively. The color of each zone represents the scale of synchronization in which colors tending to red represent higher mean synchronization and colors tending to blue represent lower mean synchronization values.

Data were submitted, in post hoc analysis, to linear mixed effects models, fit by maximum likelihood estimation, using team, role and lateral and longitudinal zones as the fixed effects and half selected as the random effect. The model also accounted for the possible interactions between all pairs of fixed effects with an exhaustive assessment of their statistical significance using suitable null models. The results from the models showed that the mean synchronization value for the home team had a $3.812\times 10^{-2}\pm 0.536\times 10^{-2}$ increase with respect to the away team ($X^{2}\left(41\right)=259.8,\;p<0.001$). Additionally, changing the role from with ball to without ball increased synchronization by $16.715\times 10^{-2}\pm 0.283\times 10^{-2}$ ($X^{2}\left(41\right)=12227.0,\;p<0.001$). The interaction between team and role effects in the home team and without ball role case led to a decrease in synchronization of $-9.178\times 10^{-2}\pm 0.629\times 10^{-2}$ ($X^{2}\left(21\right)=3713.0,\;p<0.001$). The impact of each zone in the synchronization values, for the without ball role case, is shown in Figure 3. The results indicated that when the ball was in the central areas, close to the goal being attacked (see attack direction), mean synchronization values decreased. The statistical significance for the lateral and longitudinal zone effects where respectively, $\left(X^{2}\left(63\right)=10993.0,\;p<0.001\right)$ and $\left(X^{2}\left(59\right)=17775.0,\;p<0.001\right).$ The results obtained in the statistical significance tests for the interaction between factors involving pitch zones are shown in Table 1.

### 3.2. Reciprocal Compensation

To measure reciprocal compensation, we obtained the relative phase between each player and the team. Reciprocal compensation indicates how teams reorganize and adapt in different events of the match and captures compensations among players within the teams. Figure 4 shows the deviation of each player from the team (Figure 4a) and how these deviations influenced the synchronization values (Figure 4b). The deviations of each player to the team were continuously varying (compensating), as seen in the top graphs (Figure 4a). However, there were critical moments, as shown by the apexes of the synchronization values (Figure 4b), when the balance seemed to brake (there was no compensation).

In the example shown in Figure 4c, three events are represented by three apexes circled in the synchrony measures of the away team (red). In the first event, the away team lost the ball, showing a moment of unbalance, represented by a drop in the values of the group synchronization level. These values ended up being compensated but in the following highlighted event (an interior pass by the home team), the away team lost balance again and the values of synchronization dropped again. In the third event (a cross from the home team to the area of the away team) there was another moment of unbalance which ended up in a goal for the home team (blue). These examples illustrate the influence that changes in each player-team relative phase have on the values of team synchrony, in specific events. Figure 4 also shows a tendency for the team to compensate after an event that generated unbalance, in which the team compensated deviations of individuals. Moreover, Figure 4a shows that there were moments when individual lines deviated from the team (away from 0) and those deviations were compensated for in the team synchrony measures by players whose relative phase tended towards 0 (in-phase with the team value).

### 3.3. Interpersonal Linkages

To analyze how each player contributed individually to the team, the relative frequency of each player PBGA was calculated for both with ball and without ball roles. Figure 5 shows an example of the PBGA relative frequency of four players of the away team, during the first half of the match.

Figure 5 shows that some players had a higher relative frequency in a specific PBGA range while others had a more varied amplitude of PGBA but lower relative frequency values. For the away team the total number of frames (at 25 Hz) in the first half, in which the match was running and the team had the ball, was 24,992 and without the ball was 16,774. Therefore, the polar graph of player 5, for example, shows that when the team did not have the ball, his PBGA was almost always close to 0 (relative frequency = ±6000) and ranged only from 0 to π/6 (30°). This means that from the total amount of time (frames) that his team did not have the ball, player 5 spent approximately 36% of the time in between 0 and 0.174 (10°). A PBGA close to 0 means that the player was in a position almost exactly between the goal and the ball.

Moreover, the PBGA of player 3 in the without ball role ranged approximately from 0 to π (180°) with the PBGA relative frequency being very low. This shows that the movements of player 3, in this role, were less restricted than those from player 5. However, the results were reversed, for these two players, in the role with the ball.

### 3.4. Degeneracy

To measure degeneracy, the different team configuration codes (TCC) displayed during the match, were captured using the three PBGA bins. For example, the TCC of the red team, on the first frame, was 6-4-1. This meant that there were 6 players in the first bin ([0, π/3]), 4 players in the second bin ([0, π/3]) and 1 player in the third bin ([2π/3, π]). The convex hull of the subgroups, created by the bins, is shown in Figure 6 for both teams, respectively, with the subgroup code inside each convex hull and the number of each player inside the red circles (left) and blue circles (right). The players’ numbers were included to enable the identification of changes between subgroups.

Figure 6 illustrates the relation between the TCC and the synchronization values, in three frames of a selected sequential play of the match. The frames on the left focused on the team with ball (red) and the ones on the right, being the same as the respective ones on the left, focused on the team without ball (blue). For example, the relation between the TCC and the synchronization values was captured by the sequence showed for the blue team, in Figure 6. The team displayed a TCC = 9-2-0, maintaining the levels of synchronization very high (around 0.9), in the first two frames but in the last frame, the configuration code changed to 2-6-3, lowering the levels of synchronization to 0.72. This change was generated by the ball movement of the red team.

Although the values of the order parameter (synchronization) were high in these examples, there were minor variations, indicating that different TCC may produce different values of team synchronization. Moreover, while there was interpersonal variability within the team, there was also adaptability to the momentary conditions of the match, beyond each player’s characteristics or roles, as players were not always in the same bin. Therefore, degeneracy shows how subgroups adapt within the team, without defining beforehand how many players and which specific players belong to each subgroup. Players’ behavior was regulated by the ball dynamics, adversaries’ moves and the position of the goal, highlighting the adaptable and flexible characteristics of this synergic property.

## 4. Discussion

The aim of this study was to explore the application of PBGA in CPA to directly measure player-environment relationships and capture the properties of team synergic behavior, using kinematic, sensor-based data.

To capture dimensional compression, we used synchronization as the order parameter that expresses team behavior. Mean synchrony values and linear mixed models were used to test the statistical significance of different factors such as team, half, role and zones in the values of synchronization. The mean synchrony results supported previous studies that used CPA in football [18,22], indicating that a football match is, predominantly, in a stable state, with players highly synchronized with each other in a team. Moreover, our results of cross effects in the linear mixed models also supported the results from López-Felip and colleagues, who showed significant effects for the interaction between role and distance from goal, with synchrony values decreasing in the quadrant closer to the goal [22]. In the present study, we added the analysis of the interactions between role and longitudinal and lateral zones, showing that synchrony measures decreased in zones both closer and central in relation to the goal being attacked.

Reciprocal compensation measures showed that high levels of synchronization are possible due to continuous adaptation. Players compensated the movement of each other, within the team, but also adjusted to the movement of the opponent players, the ball and the goal. Reciprocal compensation also showed that specific events in the match can unbalance the teams, which might result in a sudden change of the match equilibrium (e.g., a goal being scored). These results highlight the importance of the process of team synchrony–asynchrony–synchrony described by [24], who used readjustment delays to measure practice effects on reciprocal compensation.

The results of interpersonal linkages showed the different actions of players, in terms of their PBGA relative frequency range, according to having or not the possession of the ball. Results showed that players were more constrained, in the measures of their PBGA relative frequency range, according to their pre-defined role (defenders or attackers). For example, a player with a defender role presented a lower relative frequency range when the team did not have the ball, which suggests that players have more restricted actions when they are acting in their pre-defined roles. A previous study in football evaluated not only individual actions but also their impact on team performance by measuring the effect of individual passing actions to investigate changes in team space control, using Voronoi cells [21]. Future applications of the PBGA can also benefit from measuring the effects of individual actions in the collective performance of the team, so conclusions can be drawn regarding the influence that specific players may have on global team performance.

Finally, degeneracy was captured by identifying different team configurations and related levels of synchronization, supporting the continuous adaptive interactions and the functional role of players within the team, which goes beyond their pre-defined roles [5]. Even though changes of TCC generated variability in the values of synchronization, indicating that players values changed among PBGA bins (subgroups), the team still maintained the high values of synchrony. Finding different team configurations, using the PBGA, supports results from studies that used hypernetworks to identify the organizational dynamics within the team at several levels [34,35,36]. The results of degeneracy measurements reinforce the results from interpersonal linkages measurements by showing that variability is important in maintaining the values of the order parameter. Moreover, although football players usually have pre-defined instructions, the player-environment link is fundamental for tactical behavior in a match.

## 5. Conclusions

The results of this study showed a high connection between players and the performance environment. Introducing the PBGA in CPA enabled the direct measurement of the player–environment link, as the measures were directly influenced by the positioning of players in relation to key ecological variables. Also, it allowed the properties of team synergic behavior to be captured coherently using the synchrony measures.

Additionally, this study enabled us to suggest that CPA, with the PBGA, may capture changes in performance outcomes, before they happen, expressed in the dynamics of the synchronization values. This is a relevant topic to be addressed in future investigations, adding to the findings that variations in synchronization indicated changes in the equilibrium of the performance of the team which can be related to changes in performance outcomes.

The high rate and high resolution of this type of kinematic data enables the creation of large datasets which can capture subtle changes in performance [1]. When collected over time, these data can feed modern tools of pattern behavior classification, such as machine learning [2]. For this, collecting longitudinal data from matches and leagues is a logical next step.

## Figures and Tables

**Figure 1 sensors-20-04990-f001:**
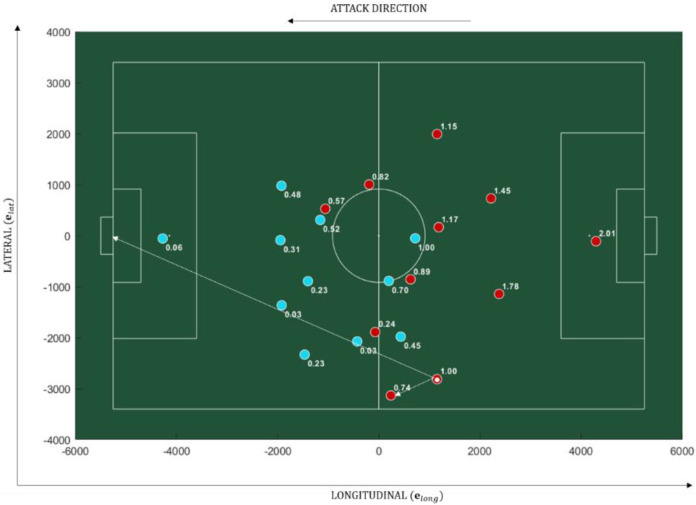
The player-ball-goal angle (PBGA). The numbers next to each player represent the PBGA in radian, measured with the vertex on the ball (white filled circle), considering the ball-player and ball-goal vectors, illustrated by the white arrows.

**Figure 2 sensors-20-04990-f002:**
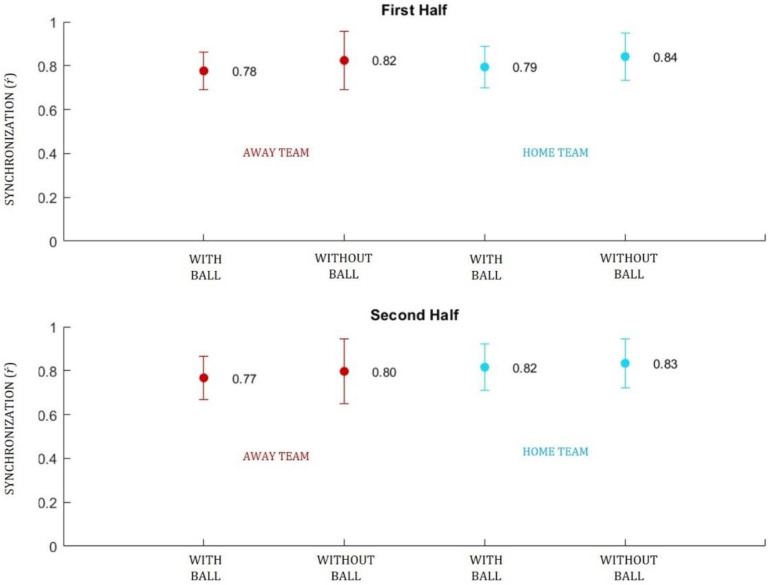
Mean and standard deviation (SD) of the synchronization ($\overset{´}{r}$) values per team, in the two halves of the match and by ball possession. Values in red represent the away team, values in blue represent the home team.

**Figure 3 sensors-20-04990-f003:**
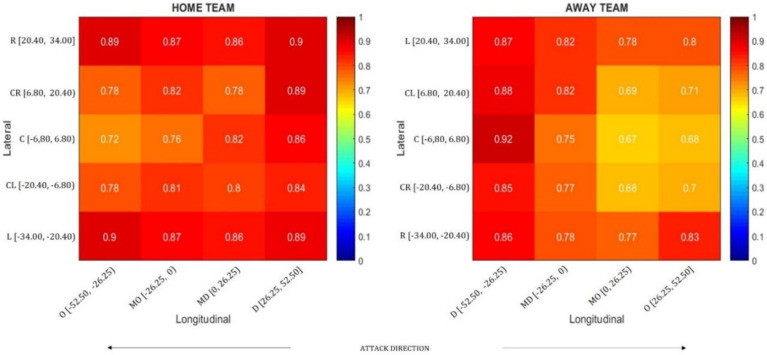
The impact of each zone in mean synchronization ($\overset{´}{r}$) values, for the without ball role. The lateral and longitudinal coordinates are represented, in meters, next to each zone code: O—offensive; MO—mid-offensive; MD—mid-defensive; D—defensive; R—right; CR—center-right; CL—center-left; L—left. The red-blue gradient scale indicates the synchronization values range from 0 to 1. Ball position data were inverted on the second half to make the results uniform for one unique direction per team.

**Figure 4 sensors-20-04990-f004:**
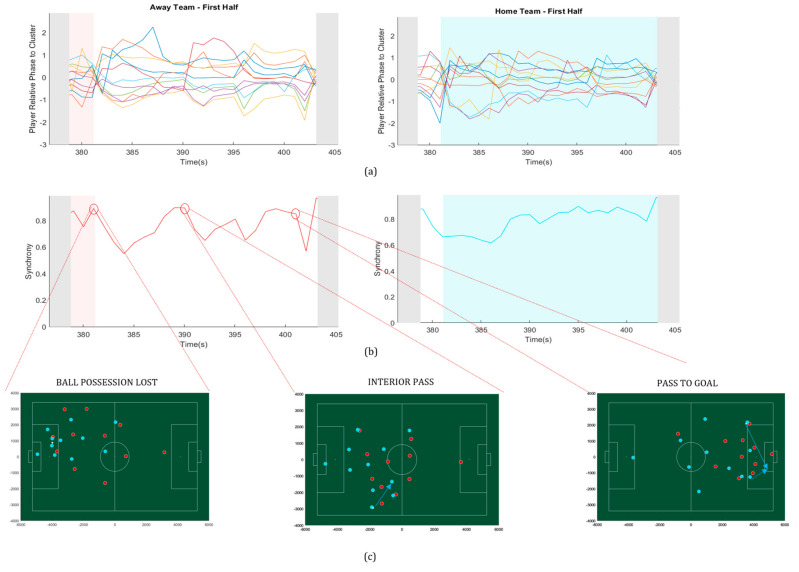
Variations of the order parameter (synchronization) and the player relative phase to the team, during a specific time frame (30 s) in the first half of the match. (**a**) Relative phase of each player to the team, in both teams; (**b**) synchrony ($\overset{´}{r}$) measures for each team. The grey background indicates that the game is not running. Red or blue background indicate when the away team (red) or the home team (blue) have possession of the ball and white background indicates that the team does not have possession of the ball; (**c**) exemplar key events of the match that are expressed by three apexes of the synchronization values, circled in red on the left-side (away team).

**Figure 5 sensors-20-04990-f005:**
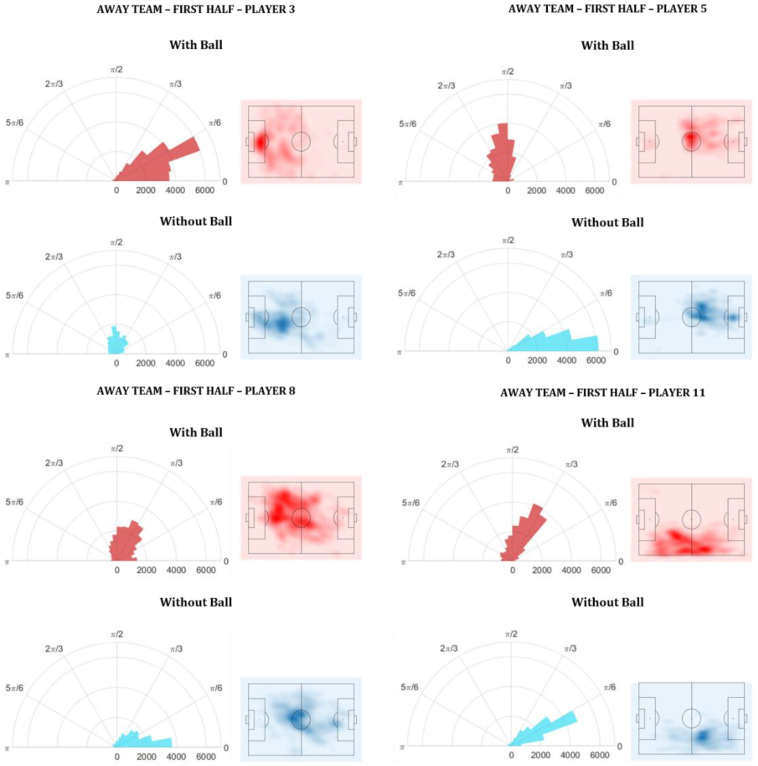
Relative frequency of the player-ball-goal angle (PBGA) illustrated by the polar graphs of four players of the away team in the roles with ball (red) and without ball (blue). The PBGA ranges from 0 to π. The relative frequency ranges from 0 to 7000 and can be seen on the bottom horizontal axis of each polar graph. To the right side of each player’s polar graph there is a heatmap of the respective player’s positions.

**Figure 6 sensors-20-04990-f006:**
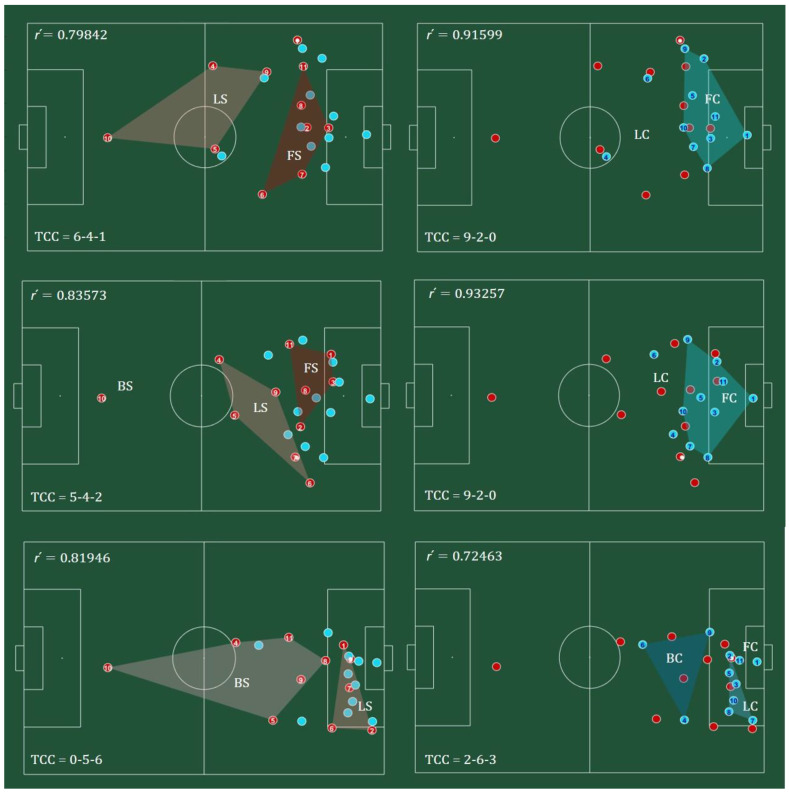
Convex hull of each subgroup in each configuration, in three selected frames of the match. Left: configurations of the team with ball (red). Right: configurations of the team without ball (blue). Subgroups are represented by the codes: FS/C—Front Support/Cover; LS/C—Lateral Support/Cover; BS/C—Back Support/Cover. On the top-left corner of each frame there is the synchronization ($\overset{´}{r}$) value and on the bottom-left corner, the team configuration code (TCC). The ball is represented by a white filled circle.

**Table 1 sensors-20-04990-t001:** Statistical significance of interaction with pitch zone effects.

Interaction	Statistical Significance
team x role	$X^{2}\left(21\right)=3173.0,\;p<0.001$
lateral zone x longitudinal zone	$X^{2}\left(48\right)=8631.8,\;p<0.001$
role x longitudinal zone	$X^{2}\left(30\right)=9886.0,\;p<0.001$
role x lateral zone	$X^{2}\left(32\right)=5662.8,\;p<0.001$
team x longitudinal zone	$X^{2}\left(30\right)=5618.3,\;p<0.001$
team x lateral zone	$X^{2}\left(32\right)=2521.3,\;p<0.001$

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
