# Peer review of "Using Optical Tracking System Data to Measure Team Synergic Behavior: Synchronization of Player-Ball-Goal Angles in a Football Match"

_sensors, 2020, doi:10.3390/s20174990_

Round 1

Reviewer 1 Report

This study aimed to test the relevance of player-ball-goal angles (PBGAs) in CPA to directly measure player-environment relationships and capture the properties of team synergic behavior using kinematic, sensor-based data. I want to congratulate authors because the high quality of the presentation of results. Moreover, the paper is interesting and novel within the tactical soccer analysis field. However, there are some major concerns, mainly in discussion section, to do before the publication of the article.

  • Abstract

I suggest to rewrite the three first sentences (lines 17-23) in one sentence.

Also, expand information with methodology and practical applications.

  • Introduction

Line 40. I suggest to include the following reference:

Clemente FM, Castillo D, Los Arcos A. Tactical analysis according to age-level groups during a 4 vs. 4 plus goalkeepers small-sided game. International Journal of Environmental Research and Public Health. 2020;17(5).

Line 40-43. The use of first person in verbal forms should not be used.

Line 88-90. The use of first person in verbal forms should not be used.

Please, link with connectors the ideas expressed in lines 87-102.

Please, link the two first paragraphs within “Cluster Phase Analysis” section.

  • Discussion

Lines 347-363. The discussion of results is poor. Authors should include at least 2-3 paragraphs discussing the results obtained in this article with the literature and postulating some practical applications.

Lines 361-375. From “In conclusion, …) one paragraph should be included.

  • Citations and references.

Authors should review the format of citations within the text and the final list of references because there several mistakes attending to the required format of the journal.

Author Response

Dear Editors and Reviewer,

We have carefully addressed one by one the comments you have kindly provided and now we think the manuscript has improved its quality. Please see in the attachment, a detailed list of how each individual comment was addressed.

Reviewer 2 Report

Overall, the concept and technique of quantifying player / team behavior during a game is interesting and contemporary. The manuscript presents an information for the reader in a way that demonstrates applicability of a technique. Although it would seem hard to replicate the work, the authors do demonstrate the technique is worth sharing with readers.

I only have minor edits to suggest:

Abstract: Provide the reader with results.

Line 38. GPS is the acronym for Global Positioning System.

Line 40. It is suggested to avoid personifying terms like “teams’” or “goals’” as this reduces readability. Instead, something like ‘… performance of the team …’

Line 54. Edit ‘data was’ to ‘data were’. Edit throughout manuscript.

Line 63. Edit dynamics’ to read dynamics.

Author Response

(The authors gave the same response as above.)

Reviewer 3 Report

General comments

The topic of Using optical tracking system data to measure team synergic behavior is really up to date and covering the large field of technical a sport science. The introduction is quite long, however the multidisciplinary approach of the article deserve all subchapters and explanations. Although The aim of the manuscript is to evaluate the accuracy of PBGAs in CPA to directly measure the player-environment link and capture the properties of team synergic behavior in football, it is hard to anticipate what has been analyses as software output and what was done directly. Also there are not explicit accuracy calculation in the results. Therefore, I suggest to clear out the in the method and result section the automatic and direct measures and then to explicitly express the accuracy.

The whole discussion is organized simply like additional comments of the results with only two already mentioned references, so it has to be fully rewrite with clear comparison to other studies.

Specific comments:

Line 29: Some keyword are already in the title, amend the key word selection.

Line 37: in association “with” football

Line 38: not just clubs but also national coaches, associations and others.

Line 41-42: This is introduction, write what is the key advantage of the system over the others, not what you further used in study.

Line 64: is there an collective behavior how to react or also pre-dined tactics which can change some behaviors.

Line 79: The synergy is one aspect of Bernstein thoughts, but there is also entropy and movement variance. Why is only synergy mentioned?

Line 143: Why you are using italics, improve format.

Line 168: do not use abbreviation for sub-title.

Line 176: Explain explicitly all signs in both equations.

Line 234: what does it mean” slightly more synchronized“. Regarding the SD in the Figure, everything seems to be with no significant difference. Do you have any minimum detectable difference value?

Line 241: Highest improvement? Is there any numerical/stat evidence of improvement? If you have SD three time higher than difference between the mean you should not use such statement.

Line 243: The figure seems to be useful, but where is the accuracy?

Line 314: describe LC, FC in the legend

Author Response

(The authors gave the same response as above.)

Round 2

Reviewer 1 Report

Great work doing the review of the manuscript